# Assessing the Detection Potential of Targeting Satellites for Global Greenhouse Gas Monitoring: Insights from TANGO Orbit Simulations

Harikrishnan Charuvil Asokan<sup>1</sup>, Jochen Landgraf<sup>2</sup>, Pepijn Veefkind<sup>3</sup>, Stijn Dellaert<sup>4</sup>, and André Butz<sup>1, 5, 6</sup>

**Correspondence:** Harikrishnan Charuvil Asokan (hari@uni-heidelberg.de)

**Abstract.** Targeting satellite observations offer a promising avenue for detecting and quantifying anthropogenic greenhouse gas (GHG) emissions from localized point sources at high spatial resolution. In this study, we assess the detection potential of the Twin ANthropogenic Greenhouse gas Observers (TANGO) satellite mission, scheduled for 2028, using orbit simulations and the TNO Global Point Source (GPS) inventory. We examine its target selection approach across three observational scenarios, Clear-Sky, Cloud-Filtered, and Cloud-Forecast, by applying two prioritization schemes (one favoring CH<sub>4</sub> point sources over CO<sub>2</sub> and the other vice versa). Results show that, under current detection limits (TDL), TANGO can detect a large fraction of major point sources, identifying ~500 targets per repeat cycle, depending on the prioritization scheme employed. However, cloud cover significantly reduces observational yield ( $\sim$ 64–68% fewer detections). Integrating a cloud-forecast-informed target selection improves the total number of detected targets by 34.6% under CO<sub>2</sub> prioritization and 22.1% under CH<sub>4</sub> prioritization compared to the cloud-filtered scenario, demonstrating the benefits of adaptive observation strategies. We also explore a hypothetical Enhanced Detection Limit (EDL) scenario, representing the potential for future satellites with improved sensitivity. While EDL extends the range of observable sources, many of these smaller emitters are associated with greater uncertainties, highlighting the importance of well-characterized retrieval precision. Finally, we discuss the potential benefits of a satellite constellation, which could enhance revisit times and observational frequency for sources of key interest. Our results demonstrate TANGO as a case study for the capabilities and challenges of next-generation targeting satellite missions, highlighting the importance of high-resolution GHG monitoring and cloud-aware adaptation for improving global emission quantification.

Copyright statement.

<sup>&</sup>lt;sup>1</sup>Institute of Environmental Physics, Heidelberg University, Heidelberg, Germany

<sup>&</sup>lt;sup>2</sup>SRON Netherlands Institute for Space Research, Leiden, the Netherlands

<sup>&</sup>lt;sup>3</sup>Royal Netherlands Meteorological Institute (KNMI), RD Satellite Observations, De Bilt, The Netherlands

<sup>&</sup>lt;sup>4</sup>Department of Air Quality and Emissions Research, TNO, Utrecht, the Netherlands

<sup>&</sup>lt;sup>5</sup>Heidelberg Center for the Environment, Heidelberg University, Heidelberg, Germany

<sup>&</sup>lt;sup>6</sup>Interdisciplinary Center for Scientific Computing, Heidelberg University, Heidelberg, Germany

# 1 Introduction

25

We are entering a climate regime where warmer conditions are becoming the norm, unprecedented in recorded human history. Anthropogenic greenhouse gas (GHG) emissions, particularly carbon dioxide (CO<sub>2</sub>) and methane (CH<sub>4</sub>), are driving this shift through their relatively long atmospheric lifetimes and strong infrared-absorbing properties, thereby amplifying radiative forcing. The latest ERC report on GHG emissions of all world countries (Crippa et al., 2023) indicated that global anthropogenic GHG emissions increased by 62% in 2022 compared to 1990 levels. Thus, tracking and recording anthropogenic CO<sub>2</sub> and CH<sub>4</sub> emissions are pivotal factors influencing international and national carbon quantification systems.

Major human-induced sources of  $CO_2$  include fossil fuel combustion, such as coal, natural gas, and oil; clearing of forests and other land use changes; and industrial processes like cement manufacturing (Friedlingstein et al., 2023). Anthropogenic sources of  $CH_4$  include activities such as cattle farming, rice cultivation, natural gas production and distribution, oil production and associated gas venting, coal mining, municipal solid waste from landfills, and wastewater treatment (Olivier, 2022). The global carbon cycle absorbs some of these emissions; however, a significant part of  $CO_2$  remains in the atmosphere, increasing concentrations (Friedlingstein et al., 2023).  $CH_4$ , while accounting for a smaller portion compared to  $CO_2$  in the atmosphere, has approximately 28–34 times more global warming potential over a 100-year period than  $CO_2$  (Myhre et al., 2013). Therefore, addressing both  $CO_2$  and  $CH_4$  emissions is vital for effectively mitigating climate change (Etminan et al., 2016).

At the forefront of counteraction strategies is the Paris Agreement, which came into effect in 2016. As of December 2024, 194 states and the European Union, representing over 98% of global GHG emissions, have signed and ratified the Agreement. The primary long-term objective of the Paris Agreement is to keep the rising global mean temperature well below 2°C above pre-industrial levels while striving to limit the rise at 1.5°C (UNFCCC, 2015). However, recent climate data indicate that 2024 is the first year where global temperatures have surpassed this 1.5°C threshold, with an annual average temperature of approximately 1.6°C above the 1850–1900 baseline (World Meteorological Organization, 2025; Copernicus Climate Change Service, 2025; McCabe, 2025). This milestone underscores the accelerating pace of global warming and places additional pressure on international climate mitigation efforts. Yet, progress toward these goals remains uncertain. Despite temporary reductions in CO<sub>2</sub> during unprecedented events like the COVID-19 pandemic (Le Quéré et al., 2020; Liu et al., 2020), achieving the Agreement's goals calls for more comprehensive global actions beyond CO<sub>2</sub> reductions. This realization led to the Global Methane Pledge at COP26 in November 2021, in which the participating parties committed to significantly reducing methane emissions (Malley et al., 2023).

To alleviate the effects of climate change due to the rise in GHG concentrations, we require reliable climate predictions. This mandates a thorough understanding of the sources and sinks of CO<sub>2</sub> and CH<sub>4</sub>, which cannot be fully addressed by the limited spatial coverage of ground-based measurement techniques, regardless of their high accuracy (Jacob et al., 2022; Zhao et al., 2023). Hence, we utilize satellite remote sensing to fill these data gaps with precise and accurate measurements of column-averaged dry air mole fraction of CO<sub>2</sub> and CH<sub>4</sub>, i.e., XCO<sub>2</sub> and XCH<sub>4</sub> (Butz et al., 2011; Jacob et al., 2022).

Satellite measurements of XCO<sub>2</sub> and XCH<sub>4</sub> can be broadly classified into two categories: mapping satellites and targeting satellites. Mapping satellites such as GOSAT, OCO-2, OCO-3, Sentinel-5 Precursor, and Sentinel 5 provide ground pixel resolution on the order of kilometres (1–10 km) and are designed to capture natural carbon fluxes on a subcontinental scale (Eldering et al., 2017; Wang et al., 2019; Lorente et al., 2021). They have broader scanning swath widths spanning from 300 to 3000 km and, under favorable conditions, can also be used to measure large emission sources (Nassar et al., 2017; Reuter et al., 2019; Pandey et al., 2019; Hakkarainen et al., 2021; Sierk et al., 2021).

To ensure accurate quantification and precise anthropogenic carbon emission reporting, we require systems that can detect and quantify relatively smaller emissions left out by mapping satellites. Targeting satellites provide a new avenue in this context due to their ground pixel resolution ranging from several tens to a few hundred meters, depending on the mission. Such high resolution enables these satellites to target smaller emission sources and allows for the visualization of plume imagery, providing emission attribution to sectors such as coal-fired power plants, fossil fuel production, large industrial facilities, and landfill sites (Qian, 2021; Guanter et al., 2021; Irakulis-Loitxate et al., 2022; Sherwin et al., 2023). Advances in satellite technology, such as the compact spectrometer concept proposed by Strandgren et al. (2020), have demonstrated the feasibility of monitoring localized CO<sub>2</sub> emissions from medium-sized power plants, a critical yet under-monitored category contributing significantly to global emissions.

Despite their capabilities, targeting satellites come with their own challenges. Their narrower swath width, ranging from 30 to 50 km, creates difficulties in achieving global coverage within a reasonable number of overpasses. Hence, these missions demand careful planning and policy for prioritizing target selection to maximize their observational capabilities. Global cloud coverage is the other major challenge, as approximately 70% of the Earth's surface is covered by clouds at any given moment (Stubenrauch et al., 2013). This is a general challenge of GHG remote sensing using satellite observations, as cloud cover significantly obstructs visibility and reduces the number of viable targets, directly affecting observational yield.

In this work, we explore the potential of targeting satellites in capturing CO<sub>2</sub> and CH<sub>4</sub> point emitters of lower emission levels. Satellites such as TANGO, GHGSAT, and CarbonMapper have the ability to maneuver and dynamically adjust observation plans as they are designed to focus on specific high-priority targets. To assess the feasibility of these systems, we simulate satellite trajectories under various scenarios to determine how many targets can be observed based on the TNO GPS inventory (Dellaert et al., 2024). Additionally, we assess the challenge of cloud cover, as it significantly reduces data yield. To address the challenge of cloud clearing, our study explores the potential benefits of integrating cloud forecast information into satellite systems to improve target detection efficiency.

The orbital parameters, detection limits, and observational capabilities of the TANGO satellite are used as a proxy to assess the performance of targeting GHG satellites. The methodologies developed in this study can be extended to or applied to similar observing systems.

The manuscript is organized as follows. Section 2 introduces the TANGO satellite mission and the global point source inventory used as a baseline dataset. Section 3 describes the simulation design and methodology. Then, in Section 4, we present the main results and findings from the simulations under different scenarios, while Section 5 discusses the implications

of these findings in the context of targeting satellite-based GHG monitoring. Finally, Section 6 concludes the manuscript with a summary of the key insights and their relevance to future satellite missions.

# 2 Satellite Parameters and Emission Inventory

This section describes the TANGO satellite mission and the emission inventory used in this study. We first introduce the satellite's capabilities, including its observational strategy and detection thresholds, followed by an overview of the Global Point Source (GPS) Inventory, which provides spatially explicit data on CO<sub>2</sub> and CH<sub>4</sub> emissions. These elements form the foundation for simulating satellite-based emission detection.

#### 2.1 TANGO Satellite Mission



The Twin ANthropogenic Greenhouse gas Observers (TANGO) is an upcoming satellite mission scheduled to launch in 2028 as part of the European Space Agency's (ESA) SCOUT program (Landgraf et al., 2024). The mission consists of two CubeSat satellites: TANGO-Carbon, which focuses on measuring CO2 and CH4, and TANGO-Nitro, which measures NO2 emissions, both targeting point source emitters such as power plants, industrial sites, and oil and gas production facilities. The temporal co-registration between CO<sub>2</sub>/CH<sub>4</sub> and NO<sub>2</sub> measurements is designed to be less than 60 seconds, enabling synchronized observations of co-located emissions from the same sources. TANGO-Carbon will measure sunlight reflected by the Earth and its atmosphere in the 1.6  $\mu$ m spectral range (1590–1675 nm) with a spectral resolution of 0.45 nm and a spectral sampling of 0.15 nm. The pushbroom spectrometer is designed to achieve a signal-to-noise ratio of 270 at the spectral continuum of a reference scene with a solar zenith angle (SZA) of 70°, a viewing zenith angle (VZA) of 0°, and a Lambertian surface albedo of 0.15 ( $L_{\rm ref}=3.16\times10^{12}\,{\rm photons\,sr^{-1}\,cm^{-2}\,nm^{-1}\,s^{-1}}$ ). This capability enables the detection of CO<sub>2</sub> sources larger than 2 Mt/year and CH<sub>4</sub> sources larger than 5 kt year<sup>-1</sup>. TANGO-Nitro will use the visible spectral range (400–500 nm band) to assist in plume detection. The satellite's narrow swath width of 30 km and ground pixel resolution of approximately 300 meters make it suitable for detecting localized emission sources. TANGO uses two agile CubeSat satellites, to be launched into a low-Earth, sun-synchronous orbit at approximately 500 km altitude. The satellites' agility allows them to dynamically adjust their observation strategy—through roll, pitch, and yaw maneuvers—enabling them to prioritize and scan high-emission targets, enhancing their versatility compared to more static satellites. Additionally, TANGO will operate in a late-morning orbit, with an equatorial crossing time of approximately 10:30, as assumed in this study. In our simulations, the TANGO-Carbon orbital parameters and detection limits are used as a proxy for the targeting satellites to evaluate their capabilities in measuring point source targets of CO<sub>2</sub> and CH<sub>4</sub> and determining how many of these sources can be detected under various scenarios.

# 2.2 Global Point Source Inventory

This study uses the TNO Global Point Source (GPS) inventory as a foundational dataset for simulating satellite target selection and emission data. The GPS inventory provides spatially explicit emission data for point sources of  $CO_2$ ,  $CH_4$ , and  $NO_x$ . It covers emissions worldwide across key sectors such as power plants, iron/steel production, cement production, refineries,

Figure 1. Global distribution of  $CO_2$  and  $CH_4$  emission sources and their cumulative emission contributions based on the GPS inventory. The left panels show the spatial distribution of  $CO_2$  (top) and  $CH_4$  (bottom) point sources, with emission rates represented by colour intensity. The right panels display the cumulative emission distributions for  $CO_2$  (top) and  $CH_4$  (bottom); vertical lines indicate the proportion of emissions covered under TDL (dashed blue lines) and EDL (dashed purple lines) relative to the point sources included in the GPS inventory.

landfills, coal mines, oil and gas production facilities, and other relevant industries. The dataset is compiled from multiple regional and global sources, including the CoCO2 2018 global point source database, Climate TRACE, and the Global Energy Monitor (GEM). Additionally, the inventory includes gridded emission information for diffuse sources, those that could not be identified as point sources, making the GPS inventory comprehensive for global emission estimates.

Given TANGO's detection limits (TDL), a subset of the GPS inventory was created for simulation purposes, containing only CO<sub>2</sub> sources with emissions larger than 2 Mt year<sup>-1</sup> and CH<sub>4</sub> sources larger than 5 kt year<sup>-1</sup>. In addition to these baseline detection limits, simulations were extended to test more optimistic thresholds, including 0.5 Mt year<sup>-1</sup> for CO<sub>2</sub> and 1 kt year<sup>-1</sup> for CH<sub>4</sub>, to evaluate the feasibility of detecting smaller emitters (Enhanced Detection Limits, EDL). These baseline inventories, derived by applying TDL and EDL thresholds to the TNO GPS dataset, form the foundation for evaluating TANGO's target selection strategies and emissions coverage, as further discussed in Sections 4 and 5.


Figure 1 illustrates the global distribution of CO<sub>2</sub> and CH<sub>4</sub> emission sources from the GPS inventory. The maps in the left panels display the spatial clustering of emission sources, while the right panels show the cumulative emission distributions for CO<sub>2</sub> and CH<sub>4</sub>. These distributions emphasize the contributions of smaller emitters to global totals. This figure parallels the cumulative emission analysis in Strandgren et al. (2020) (Figure 1), which focused on power plant emissions using similar methods. Figure 1 expands on this by including CH<sub>4</sub> sources alongside CO<sub>2</sub> and covering a broader range of industrial sectors. TANGO's current detection limits (TDL) (dashed blue lines) capture 54.3% of CO<sub>2</sub> emissions and 78% of CH<sub>4</sub> emissions relative to the large (industrial) point sources included in the GPS inventory, whereas the Enhanced Detection Limits (EDL) (dashed purple lines) increase the coverage to 89.5% for CO<sub>2</sub> and 91.6% for CH<sub>4</sub>. It should be noted that the GPS inventory, while comprehensive for industrial point sources, does not cover all anthropogenic emissions. Emissions from sectors such as road transport, residential buildings, and agriculture (for CH<sub>4</sub>) are represented as gridded diffuse emissions, which were not included in this study as we focus exclusively on point sources. According to Dellaert et al. (2024), the total TNO GPS dataset accounts for 23% of global CH<sub>4</sub> emissions and 47% of global CO<sub>2</sub> emissions relative to total anthropogenic emissions as reported in the EDGAR dataset (Crippa et al., 2023).





An overview of the spatial distribution of TDL and EDL point sources across continents is shown in Figure 2. Under TDL thresholds, the inventory includes significant emitters across all continents. For instance, in Asia, 1114 CO<sub>2</sub> sources account for 4623.76 Mt year<sup>-1</sup> of regional emissions, while in North America, 652 CH<sub>4</sub> sources contribute 5215.88 kt year<sup>-1</sup>. These numbers reflect the major emitters detectable within TDL, offering insights into the satellite's immediate operational capabilities.

Distinct regional patterns also emerge in the emission inventory. Asia's dominance in CO<sub>2</sub> and CH<sub>4</sub> emissions reflects its industrial density and energy demands, whereas North America's high CH<sub>4</sub> emissions are driven by its extensive oil and gas infrastructure. Although Africa and Oceania host fewer emitters within the thresholds, their contribution to total emissions remains noteworthy.

The non-linear relationship between point source counts and cumulative emissions in Figure 2 underscores the value of improved detection limits for both CH<sub>4</sub> and CO<sub>2</sub>. For CH<sub>4</sub>, increasing sensitivity from TDL to EDL expands the inventory from 4035 to 11,897 point sources (a 194.8% increase), yet the cumulative emissions increase by only 17.5%. This pattern indicates that many additional sources included under EDL are smaller emitters contributing less individually. A similar trend is observed for CO<sub>2</sub>, emphasizing the importance of incorporating smaller emitters into emission inventories for future missions to achieve comprehensive monitoring.

While EDL reflects a broader capability beyond TANGO's immediate scope, these insights align with the mission's objective of advancing GHG monitoring technologies. The results serve as a stepping stone for developing similar missions to improve global emissions inventories and address limitations inherent to current systems.

Figure 2. Continental distribution of point sources based on the GPS inventory under TDL and EDL. Bar plots show point source counts for  $CH_4$  (sky blue) and  $CO_2$  (orange) under EDL, with striped patterns representing TDL overlaid on EDL bars. Meanwhile, the line plots overlay cumulative emissions as a function of these thresholds, with dashed lines representing EDL and solid lines representing TDL. The line plots are colour-coordinated with their respective y-axes, with sky blue lines and labels representing  $CH_4$  emissions (kt year<sup>-1</sup>) and orange lines and labels representing  $CO_2$  emissions (Mt year<sup>-1</sup>).

## 3 Simulation Design and Methodology

In this section, we outline our simulation workflow. We start from the TNO Global Point Source inventory to list all  $CO_2$  and  $CH_4$  emitters. Next, we define observation schemes based on two detection thresholds (TDL and EDL) and two prioritization rules: one favoring  $CO_2$  sources and one favoring  $CH_4$  sources. Then, for each scheme, we run a four-day orbit simulation that includes TANGO's roll and pitch maneuvers and excludes overpasses with solar zenith angles above  $70^\circ$ . After that, we apply three cloud-treatment approaches: ideal clear-sky conditions, post-selection filtering using MODIS cloud masks, and pre-selection using cloud forecasts (with a one-day offset test). Finally, we compare, analyze, and discuss the detected targets under each scenario.

# 165 3.1 Satellite Trajectory Simulation


In this study, we employed an orbit simulator to generate satellite trajectories for TANGO. Given the satellite's orbital parameters, a four-day simulation period was selected, as TANGO requires approximately four days to complete one full orbit cycle and return to the same geographical position (repeat cycle). The simulation replicated TANGO's orbital motion at an altitude of 500 km, with a local time of ascending node (LTAN) of 10:30 hours and a near-circular orbit (eccentricity = 0.0).

**Figure 3.** Schematic Illustration of TANGO's dynamic maneuverability using roll and pitch adjustments during an observational cycle. The plot represents a 10° roll maneuver, showing key stages of satellite operation: reorientation, stabilization, and imaging.

We conducted the simulations for the year 2022 with different periods of four consecutive days to study the seasonal performance dependencies of the mission. Each simulation run started at a different epoch, covering various dates throughout the year. This approach allowed us to simulate how the satellite would pass over all continents during different seasons, accounting for daylight conditions to optimize point source detections. The TANGO mission commits to data quality for solar zenith angles (SZA) below 70°, and therefore we excluded overpasses with larger SZA in the post-simulation data filtering, as these conditions are generally unfavorable for imaging due to increased atmospheric path lengths and lower signal-to-noise ratios. It is important to note that the actual orbital parameters of the TANGO mission may vary, but these simulated trajectories provide a good approximation for our purpose of understanding the emission target detection capabilities of targeting satellites.

# 3.2 Satellite maneuverability and Target prioritization strategy




To overcome the challenge of the narrow swath width inherent to targeting satellites like TANGO, the satellite system is designed to perform dynamic adjustments using its roll, pitch, and yaw angles. This maneuverability allows the satellite to detect targets within a 30° roll on either side of its orbital path, extending its coverage beyond the narrow nadir-viewing scenario. For example, during its orbital motion, based on a predefined target list, if a target is located 10° to the left of the orbital path, the satellite will reorient itself to capture the target and then return to its original position. However, other potential targets in the vicinity might not be seen during operations as the satellite adheres to a predefined prioritization policy.

Several factors influence this satellite maneuver procedure. Once a target is selected, the satellite requires time for reorientation to that specific angle, followed by a stabilization period to ensure the system is ready for scanning. The satellite then has a limited imaging window, during which data is recorded. Subsequently, the satellite reverts to its original nadir-viewing trajectory. Depending on the roll angle and the satellite's hardware capabilities, this maneuver takes several seconds. Figure 3 provides a schematic representation of the satellite's maneuvering process.

In this study, we simulated a prioritization policy to address the clustering of point source targets and to exhibit the effects of these maneuvers. Many point source emitters tend to cluster in specific regions due to the concentration of industrial,

agricultural, and fossil fuel-related activities or appear within a short time window during the satellite's overpass. In such cases, where multiple targets are detected in proximity, targets are selected for scanning based on a predefined prioritization policy. For this study, we used two distinct prioritization strategies: one prioritizing large CO<sub>2</sub> emitters and switching to CH<sub>4</sub> emitters when no significant CO<sub>2</sub> sources were available, and the other focusing on large CH<sub>4</sub> emitters and switching to CO<sub>2</sub> targets under similar conditions. These strategies were tested across three scenarios—Clear-Sky, Cloud-Filtered, and Cloud-Forecast—to evaluate the satellite's detection capabilities under varying conditions. (These scenarios are described in detail in Sections 3.3 and 3.4). To simulate these scenarios, we explicitly modeled the satellite maneuvers required to target emissions. The maneuvers were calculated based on the angular location of each target relative to the satellite's orbital path. This approach ensured that the simulations accounted for realistic constraints, such as the time required for reorientation and stabilization.

# 3.3 Cloud Filtering and Target Optimization






Using the processes described in Sections 3.1 and 3.2, we simulated ideal target selection, referred to here as "Clear-Sky conditions." In this scenario, all potential emission sources of CO<sub>2</sub> and CH<sub>4</sub> are assumed to be fully observable without interference. However, while the Clear-Sky scenario represents the theoretical maximum detection capacity of the satellite within the constraints described in the sections above, it does not account for the significant impact that global cloud coverage has on real-world observations. Hence, to simulate more realistic operational conditions, we incorporated a cloud filtration strategy into our workflow. We utilized satellite-based cloud mask data from the MOD35\_L2 product (DOI: 10.5067/MODIS/MOD35\_L2.061), derived from the Moderate Resolution Imaging Spectroradiometer (MODIS) aboard NASA's Terra satellite (Ackerman et al., 2015). The MODIS cloud mask provides confidence levels for each pixel, categorized as cloudy, uncertain, probably clear, or confidently clear. Data from individual MODIS granules, representing 5-minute orbital segments, were reprojected onto a uniform global grid. To reduce data gaps and ensure consistent spatial coverage, we averaged the cloud confidence values over four days, corresponding to the satellite's repeat cycle. Only grid cells that remained entirely invalid (e.g. missing data) over the four days were excluded from further analysis. By incorporating this process, we reduced the rejection of cloud data, ensuring that observations from partially clear regions contributed to the analysis, thereby improving data retention for target detection under realistic cloud conditions.

The cloud filtration process involves two steps, as depicted in Figure 4. After selecting the targets, in the first step, we assess the cloud coverage within a 50 km radius of each emission source. If the cloud cover exceeds 70%, the target is flagged as unsuitable for observation and excluded from further analysis. This coarse-scale screen balances discarding hopelessly overcast regions against retaining potentially observable scenes. Krijger et al. (2007) show that allowing up to 20% cloud at footprints of  $\sim$ 10000 km² raises clear-scene yield from  $\sim$ 3% (zero-cloud requirement) to  $\sim$ 17%. By analogy, our 70% cloud cut at 50 km excludes only the worst 30% of overcast areas while keeping most viable targets. For targets with less than 70% cloud cover, we proceed to the second filtering step: a localized 3 km-radius check that retains only those with  $\leq$  30% cloud cover (i.e.,  $\geq$  70% clear pixels). This finer check leverages TANGO's  $\sim$ 300 m pixels to exploit narrow "windows" through broken clouds. Frankenberg et al. (2024) demonstrate that  $\sim$ 200 m retrievals boost clear-scene frequency by 5–10× compared to kilometer-scale footprints. At 300 m resolution, a 3 km circle covers roughly 300 independent pixels, so requiring  $\geq$  70% clear leaves

**Figure 4.** The schematic flowchart shows the logical sequence of the two-step cloud filtering process applied to emission target selection. The diamonds represent decision-making points, circles denote specific conditions to be evaluated, such as cloud coverage thresholds within a 50 km or 3 km radius of the target, and squares indicate outcomes or actions.

more than 210 valid soundings per target. While this does not imply that retrievals can consistently produce a complete data product under such conditions, it provides sufficient clear pixels for robust plume detection. Only targets that pass both steps are considered viable, ensuring a high probability of visibility. These thresholds were chosen based on insights from the cited studies for simulation purposes and may not universally apply to all regions or cloud types, as persistent cloud formations, especially in the tropics, might require further refinement. By implementing this filtering mechanism, we better understand the influence of cloud cover and optimize the selection of realistic target numbers.

By integrating this cloud-clearing methodology, we optimize target selection, offering insights into the influence of cloud cover on satellite performance and enhancing the overall reliability of our simulations. These results provide a more realistic understanding of how many emission sources can be detected compared to the idealized clear-sky conditions.

# 3.4 Cloud-Forecast integration



To improve the detection of targets hindered by global cloud cover, we propose integrating high-resolution cloud forecast information into the satellite systems. Operationally, we assume forecast fields for the upcoming cycle are available 24 h in advance, with daily updates to the target list. In the second scenario (Cloud-Filtered), targets are selected solely based on the prioritization strategy using emission levels without considering the likelihood of clear skies near the targets. These targets are then filtered using a two-step cloud filtration process (see Section 3.3), removing all targets within cloud-covered regions. While this ensures that the remaining targets have a higher probability of visibility, the cloud-clearing process significantly

reduces the number of usable targets for the satellite. However, if forecast information on cloudiness is available, observations could be redirected to alternative targets that are more likely to be clear of clouds, enhancing the overall data yield. The third scenario (Cloud-Forecast) builds on this concept by integrating cloud forecasts into the target selection process.

In the Cloud-Forecast scenario, we assume perfect forecast data availability, using MODIS cloud mask data (MOD35\_L2 product) as a proxy for high-resolution cloud forecasts. Simulations begin by pre-filtering the GPS inventory through the two-step cloud filtration process, based on forecast data specific to each four-day simulation cycle, to remove sources likely obscured by clouds. This results in a refined inventory of emission sources with high visibility potential for each cycle. Dynamic maneuvering and cluster-based prioritization strategies are then applied to this filtered inventory, incorporating forecast data to guide the satellite's trajectory. By identifying and prioritizing alternative targets with better viewing conditions, the satellite bypasses cloud-covered regions and focuses on observable areas, improving detection efficiency. Since this scenario assumes no forecast errors, additional post-selection filtering based on actual cloud conditions is not required.

To evaluate the effects of forecast inaccuracies, we introduced a subcase called the "Forecast +1d" scenario. Here, the MODIS cloud mask data for one four-day period serves as the forecast, while the actual cloud conditions are represented by a four-day average shifted by one day. Targets are first pre-filtered using the forecast data to create the refined inventory and plan satellite maneuvers. Post-maneuver, the actual cloud data is applied to filter out any remaining cloud-covered targets, accounting for errors introduced by the temporal misalignment. This setup represents a simplified model of forecast errors, evaluating how such inaccuracies might affect target detection. While the Cloud-Forecast scenario assumes perfect forecast accuracy, the Forecast +1d scenario provides a baseline to estimate detection variability under imperfect conditions. These scenarios offer a systematic approach to assess how integrating cloud forecasts and accounting for forecast inaccuracies impact the operational efficiency and detection potential of targeting satellite missions.

# 4 Results







In our analysis, we evaluated the detection potential of the TANGO satellite under three simulation scenarios: Clear-Sky, Cloud-Filtered, and Cloud-Forecast. These scenarios examine variations in detection performance under ideal and near-realistic conditions.

#### 4.1 Evaluation of Simulation Scenarios

The Clear-Sky scenario simulates ideal conditions, with all emission sources observable without cloud interference, providing an upper limit of the satellite's detection potential. This case allows us to evaluate the maximum target coverage within TANGO's operational constraints, establishing the baseline detection capacity for CO<sub>2</sub> and CH<sub>4</sub> point sources. In contrast, the Cloud-Filtered scenario incorporates realistic global cloud coverage by applying a two-step cloud filtration strategy. This scenario emphasizes the impact of cloud cover on the satellite's ability to detect emission sources. Lastly, the Cloud-Forecast scenario demonstrates potential improvements by integrating cloud forecast data to guide satellite operations.

Figure 5 compares the three simulation scenarios: Clear-Sky (top panel), Cloud-Filtered (middle panel), and Cloud-Forecast (bottom panel) conditions, based on a four-day simulation period from March 18–22, 2022, representing a northern hemisphere spring. The light blue bands on the map indicate the width of the satellite's trajectory, encompassing TANGO's maximum roll angle capability, which defines the spatial extent within which targets can be detected.

In each panel, detected targets are colour-coded by emission sector, with CO<sub>2</sub> sources represented by circles and CH<sub>4</sub> sources by squares. The figure illustrates the CO<sub>2</sub> prioritization strategy, where priority is based on emission size, and when CO<sub>2</sub> sources are unavailable, large CH<sub>4</sub> sources are selected. The middle and bottom panels overlay cloud coverage derived from MODIS cloud mask data and qualitatively exhibit the impact of cloud cover on target detection. During this four-day simulation period, the Clear-Sky scenario identifies 435 targets, which are reduced to 120 in the Cloud-Filtered scenario due to cloud interference. Incorporating cloud forecasts in the Cloud-Forecast scenario improves this detection count to 177. This visualization highlights the reduction in detectable targets under cloud-covered conditions and demonstrates the improvements achieved by cloud-forecast integration, which helps circumvent cloud-filled areas.

# 4.2 Monthly Detection Patterns Across Scenarios








To quantitatively assess target detection, we conducted simulations for the entire year of 2022, generating target counts every four days. Figures 6 and 7 show the monthly average number of targets detected under each scenario, calculated by averaging all four-day results within each month. Error bars represent the standard deviation, indicating variability around these monthly averages. Figure 6 illustrates selected targets under the CO<sub>2</sub> prioritization scheme for TDL, while Figure 7 depicts results for the CH<sub>4</sub> prioritization scheme for both TDL and EDL. In both figures, total counts represent the combined number of detected CO<sub>2</sub> and CH<sub>4</sub> sources.

Even under the same detection threshold (TDL), the total number of targets detected differs between the two prioritization strategies. Prioritization schemes implemented in this study influence both the relative balance of selected targets (CO<sub>2</sub> versus CH<sub>4</sub>) and the total number of detections. This variation arises from the inventory composition. For instance, the baseline inventory includes 4035 CH<sub>4</sub> sources compared to 1834 CO<sub>2</sub> sources under TDL, and 11,897 CH<sub>4</sub> sources compared to 6766 CO<sub>2</sub> sources under EDL. This disparity explains the higher total detections under the CH<sub>4</sub> prioritization scheme, as the larger pool of available CH<sub>4</sub> sources leads to more selections. Conversely, CO<sub>2</sub> prioritization results in fewer total detections due to the smaller pool of CO<sub>2</sub> sources.

In Figure 6, the blue line represents the Clear-Sky scenario, the green line shows the Cloud-Filtered case, and the orange line indicates the Cloud-Forecast scenario. This figure illustrates the substantial impact of global cloud cover on detection capabilities. After applying the two-step cloud filtering, the number of detectable targets is notably reduced. The yearly average target counts in the Clear-Sky, Cloud-Filtered, and Cloud-Forecast scenarios are  $404 \pm 6$ ,  $128 \pm 2$ , and  $195 \pm 3$ , respectively. This represents a  $68\% \pm 1\%$  decrease in detectable targets from Clear-Sky to Cloud-Filtered, with the uncertainty propagated from the standard errors of the mean target counts in both scenarios. This reduction is consistent with global cloud cover estimates, which indicate that approximately 70% of Earth's surface is obscured by clouds at any given time, underscoring the critical role of cloud cover in influencing satellite measurements (Stubenrauch et al., 2013).

**Figure 5.** Global distribution of detected CO<sub>2</sub> and CH<sub>4</sub> point sources based on a four-day simulation period from March 18–22, 2022, under three scenarios: Clear-Sky (top panel), Cloud-Filtered (middle panel), and Cloud-Forecast (bottom panel) for TDL simulations. Detected targets are colour-coded by emission sector, with CO<sub>2</sub> sources represented by circles and CH<sub>4</sub> sources by squares. Marker size is scaled to emission strength within each gas type. Overlaid cloud coverage is categorized as confident clear, uncertain clear, probably cloudy, and confident cloudy, based on MODIS cloud mask data. The light blue bands indicate the width of the satellite's trajectory, encompassing TANGO's maximum roll angle.

In contrast, the Cloud-Forecast scenario shows a substantial improvement in detection capability, with a  $53\% \pm 4\%$  increase in detectable targets over the Cloud-Filtered scenario. This improvement highlights the potential of near-real-time cloud fore-

**Figure 6.** Monthly average number of detected emission targets per repeat cycle for CO<sub>2</sub> prioritization under TDL (2022). The figure illustrates the total number of detected targets for each month across four scenarios: Clear Sky (sky blue), Cloud-Filtered (green), Cloud Forecast (orange), and Forecast +1d (purple). Data points reflect the total averages for CO<sub>2</sub> and CH<sub>4</sub> detections combined, with error bars indicating standard deviations within each scenario.

cast integration in boosting detection efficiency for satellite missions like TANGO, enabling more accurate and adaptive target selection even under partially cloudy conditions.




To further simulate realistic operational constraints, we evaluated the 'Forecast +1d' scenario. This subcase shifts the cloud forecast data by one day, leveraging MODIS cloud mask data averaged over four days, to model the potential impacts of forecast inaccuracies. While this approach assumes temporal misalignment, it does not represent a true forecast error scenario but is a conservative estimate to assess detection variability under operational constraints. The agreement between forecast and Forecast +1d categorical cloud mask data over the year was  $76\% \pm 1\%$ , reflecting the impact of the averaging and temporal shift used in the simulation. The Forecast +1d scenario, represented by purple dashed lines in Figures 6 and 7, illustrates the impact of forecast error on the target detection count across each prioritization scheme and detection limit setting.

In Figure 6, for the  $CO_2$  prioritization scheme, the yearly average target count in the Forecast +1d scenario was  $153\pm3$ . While this is  $22\%\pm2\%$  lower than the Cloud-Forecast scenario ( $195\pm3$ ), it remains significantly higher than the Cloud-Filtered case ( $128\pm2$ ), representing a  $20\%\pm3\%$  improvement. Despite a forecast accuracy of approximately 76%, these results demonstrate that integrating forecast data yields a tangible benefit in enhancing target detection. This finding shows the value of real-time cloud forecast integration in boosting detection efficiency for satellite missions like TANGO, allowing for more adaptive target selection even under suboptimal forecast conditions.

Figure 7 presents the detection outcomes under the CH<sub>4</sub> prioritization scheme, comparing target counts for TANGO's current detection limits (TDL) with a lower detection threshold (EDL) representative of a future satellite scenario. Similar to Figure 6, the scenarios—Clear-Sky, Cloud-Filtered, Cloud-Forecast, and Forecast +1d—are shown with different coloured lines, where dashed lines represent TDL and solid lines indicate EDL.

**Figure 7.** Monthly average number of detected targets per repeat cycle under TDL and EDL with CH<sub>4</sub> prioritization across different simulation scenarios: Clear-Sky (sky blue), Cloud-Filtered (green), Cloud-Forecast (orange), and Forecast +1d (purple). The average number of detected emission targets is shown on the y-axis, with months on the x-axis. TDL results are represented by solid lines, while EDL results are depicted with dashed lines.

In the Clear-Sky scenario, the  $CH_4$  prioritization scheme yielded a yearly average target count of  $582 \pm 7$  for TANGO detection limits, which increased to  $1157 \pm 15$  under the EDL. This substantial increase of  $99\% \pm 4\%$  accentuates the potential advantage of improved detection limits, enabling the satellite to capture smaller emitters that are undetectable with current TANGO capabilities.





The Cloud-Filtered scenario, reflecting realistic cloud cover constraints, shows considerable reductions in target counts, with a yearly average of  $212 \pm 4$  for TDL. The EDL partially offsets this reduction, raising the average count to  $398 \pm 6$ , an  $88\% \pm 5\%$  increase over the TDL within the same scenario. This increase reflects the larger pool of detectable targets under the EDL inventory, which includes smaller emitters that TDL is not designed to capture. However, TDL effectively targets major emitters, fulfilling its current operational objectives. While cloud cover imposes similar constraints on both thresholds, the enhanced sensitivity of EDL allows it to detect additional targets in clear-sky regions, translating to relatively higher numbers under cloud-filtered conditions.

Integrating cloud forecast data in the Cloud-Forecast scenario further improves detection outcomes relative to the Cloud-Filtered scenario. For TDL, the Cloud-Forecast scenario achieves a yearly average of  $272 \pm 5$  targets, marking a  $28\% \pm 3\%$  increase over the Cloud-Filtered case. Under EDL, the Cloud-Forecast scenario reaches  $527 \pm 7$ , representing a  $32\% \pm 3\%$  improvement over the Cloud-Filtered case. These results underscore the combined benefit of cloud forecast integration and enhanced detection sensitivity, which together optimize observational yield even under variable cloud conditions. For TDL, the Forecast +1d scenario yields a yearly average target of  $222 \pm 5$ , a 4.8% increase over the Cloud-Filtered case. With EDL, the improvement is more pronounced, with a 7.5% increase, from  $398 \pm 6$  in the Cloud-Filtered to  $428 \pm 6$ . These results demonstrate that even under a conservative estimate of forecast inaccuracy, integrating cloud forecast data enhances the detection capabilities, maintaining resilience in achieving target detection under suboptimal conditions.

**Figure 8.** Average number of CH<sub>4</sub> and CO<sub>2</sub> targets detected per four-day interval across continents under CH<sub>4</sub> prioritization for TDL and EDL detection limits. Detection scenarios include Clear Sky, Cloud-Filtered, Cloud Forecast, and Forecast +1d, as indicated by colour-coded markers. Solid bars represent CH<sub>4</sub> targets, while striped bars represent CO<sub>2</sub> targets for TDL. Scatter squares denote EDL detections using the same colour scheme: solid colours for CH<sub>4</sub> and striped patterns for CO<sub>2</sub>.

# 4.3 Continental Detection Distribution





Figure 8 presents the continental distribution of selected CH<sub>4</sub> and CO<sub>2</sub> targets under the CH<sub>4</sub> prioritization scheme. TDL-selected targets are displayed as bar plots, with solid colours for CH<sub>4</sub> and striped patterns for CO<sub>2</sub>. EDL-selected targets are shown as overlaid scatter markers, solid squares for CH<sub>4</sub> and striped for CO<sub>2</sub>, illustrating a possible future capability. The figure highlights how targets are distributed across scenarios and continents, focusing on the operational constraints and performance of TANGO under TDL. A numerical breakdown of the detected targets under both CH<sub>4</sub> and CO<sub>2</sub> prioritization schemes is provided in Appendix A (Tables A1 & A2), summarizing the average number of detected sources per 4-day cycle across continents for different cloud scenarios.

Compared to the baseline distribution in Figure 2, TANGO effectively selects high-emission targets under operational constraints such as maneuvering and prioritization. For instance, in Asia, the baseline TDL inventory includes 2431 CH<sub>4</sub> sources and 1114 CO<sub>2</sub> sources per 4-day interval. From this pool, an average of 206 CH<sub>4</sub> and 103 CO<sub>2</sub> targets are selected in the Clear-Sky scenario. Similarly, North America contributes 59 CH<sub>4</sub> and 14 CO<sub>2</sub> targets from its baseline pool of 652 CH<sub>4</sub> and 341 CO<sub>2</sub> sources. These results demonstrate TANGO's focus on significant emitters, enabling efficient use of its limited observational capacity.

The Cloud-Filtered scenario highlights the challenges posed by cloud cover in target selection. For example, under TDL, Europe sees 10.8 CH<sub>4</sub> targets per 4-day interval compared to 43.3 in Clear-Sky, while North America records 21.6 CH<sub>4</sub> targets compared to the Clear-Sky scenario of 58.9. However, the Cloud-Forecast scenario demonstrates TANGO's adaptability, with forecast data enabling the selection of 28.5 CH<sub>4</sub> targets in North America, up from 21.6 in Cloud-Filtered. These improvements emphasize the value of real-time forecasting in overcoming observational limitations imposed by cloud cover.

The Forecast +1d scenario, incorporating forecast inaccuracies, surpasses Cloud-Filtered conditions. For instance, under TDL, Asia records 94 CH<sub>4</sub> targets per 4-day interval, compared to 111.6 in Cloud-Forecast and 89.1 in Cloud-Filtered. Similarly, Europe retains 8 CO<sub>2</sub> targets in Forecast +1d, exceeding the 7.3 observed in Cloud-Filtered. These results demonstrate TANGO's resilience in maintaining robust target selection capabilities despite forecast variability.

The EDL results, shown as a future capability, highlight the additional detection potential for smaller emitters. For instance, in Africa under Clear-Sky, the average number of CH<sub>4</sub> targets increases from 62 (TDL) to 92.5 (EDL). Similarly, in Asia, CH<sub>4</sub> targets rise from 206.1 to 303. However, these enhancements represent a hypothetical scenario and are presented as a supplementary case to demonstrate how advanced detection thresholds could expand the satellite's observational scope.

# 375 5 Discussions





The objective of this study is to verify the capabilities of targeting satellites in detecting  $CO_2$  and  $CH_4$  point source emissions. By providing precise observations of individual emitters, targeting satellites complement existing satellite systems and fill critical gaps in emission inventories. This is pivotal in the context of global climate agreements, where monitoring smaller and distributed emitters can contribute to more comprehensive carbon budgets.

In our work, TANGO was chosen as the model satellite for simulations, as it represents a near-future realization. Our findings demonstrate the major role TANGO could play in advancing satellite-based GHG monitoring. By utilizing its Detection Limits (TDL), the satellite captures major emitters effectively under operational constraints, as evidenced by the scenarios analyzed in this study. Here, we contemplate the results of our simulations and assess their implications.

The simulations were conducted using two prioritization strategies: CO<sub>2</sub> prioritization and CH<sub>4</sub> prioritization. These strategies were chosen specifically to explore TANGO's capability in dynamically focusing on different types of emitters and do not represent the actual operational prioritization strategies. TANGO's design as a flexible research satellite allows for the development of user-defined prioritization schemes tailored to specific mission objectives. For instance, while this study prioritized high-emission sources to demonstrate detection potential, alternative strategies could target medium emitters, prioritize undermonitored sectors (e.g., agriculture, small-scale industrial facilities), or address emissions in regions with sparse ground-based observations. This adaptability makes TANGO a versatile tool for addressing diverse emission monitoring needs.

For this proof-of-concept study, the CO<sub>2</sub> and CH<sub>4</sub> prioritization schemes served as a baseline to evaluate TANGO's detection capabilities. We explored three scenarios to assess performance under varying conditions: Clear-Sky (ideal upper limit), Cloud-Filtered (realistic global cloud cover), and Cloud-Forecast (which includes a Forecast +1d subcase that incorporates forecast inaccuracies). The Forecast +1d scenario thus represents a conservative estimate designed to explore detection variability under worst-case forecast conditions. These scenarios confirm the potential of using cloud forecasts to improve detection efficiency by focusing on high-visibility regions.

We utilized the TNO global point source inventory to establish a baseline dataset for our simulations under TDL and EDL thresholds. Under TDL, the inventory contained 4035 CH<sub>4</sub> sources and 1834 CO<sub>2</sub> sources, while EDL expanded the dataset

to 11897 CH<sub>4</sub> and 6766 CO<sub>2</sub> sources. These baseline datasets served as the global pool from which targets were dynamically selected in the respective simulations, aligning with the detection limits and prioritization strategies defined in this study.







As illustrated in the cumulative emission distributions (Figure 1), TDL covers 54.3% of global CO<sub>2</sub> emissions and 78% of global CH<sub>4</sub> emissions relative to the point sources included in the GPS inventory. This substantial coverage validates TANGO's capacity to optimally target major emitters, even under its current detection thresholds. While the EDL provides more extensive coverage, raising these percentages to 89.5% for CO<sub>2</sub> and 91.6% for CH<sub>4</sub>, the contributions achieved under TDL maintain their importance in improving global emissions monitoring.

In the TDL simulations, the yearly average number of detected targets per repeat cycle (four-day interval) varies depending on the prioritization strategy applied. Under the Clear-Sky scenario, the satellite detected  $404 \pm 6$  targets when prioritizing  $CO_2$  and  $582 \pm 7$  for  $CH_4$  prioritization, representing the upper limit of detection. Considering the TDL inventory of 5869 total point sources, this corresponds to approximately 6.8% and 9.9% detection efficiency for  $CO_2$  and  $CH_4$  prioritization, respectively. This shows the satellite's capacity to capture a meaningful subset of high-priority emitters within a single repeat cycle under operational constraints.

With multiple repeat cycles, a majority of the TDL inventory could theoretically be observed if the satellite prioritized all point sources equally. However, operational constraints, such as the need for frequent revisits to certain under-monitored targets, particularly those located in regions with sparse ground-based observations or targets of emerging concern, may alter this distribution. In such cases, prioritization strategies would need to focus on repeatedly targeting these particular emitters, likely decreasing the number of unique sources observed per cycle. This illustrates a trade-off between broad coverage and targeted monitoring, showing the importance of user-defined prioritization strategies to align with specific mission objectives.

One potential refinement for future prioritization schemes could involve coupling target selection with uncertainty metrics in the GPS inventory. Sources with larger uncertainties could be assigned higher priorities, ensuring that the satellite observations reduce uncertainty in emission estimates over time. This coupling could also be implemented dynamically, allowing priorities to evolve as uncertainties are reduced through repeated observations. While this approach improves mission efficiency, it would require well-characterized and regularly updated uncertainty metrics. As such, fine-tuning prioritization schemes remains an essential step in optimizing TANGO's mission operations over the coming years.

Further, in the TDL simulations, for the Cloud-Filtered scenario, the numbers decreased to  $128 \pm 2$  for  $CO_2$  prioritization ( $212 \pm 4$  for  $CH_4$  prioritization) when cloud conditions were considered. The integration of forecast data in the Cloud-Forecast scenario improved detection, with an average of  $195 \pm 3$  targets under  $CO_2$  prioritization ( $272 \pm 5$  for  $CH_4$  prioritization). Finally, in the Forecast +1d subset, where forecast inaccuracies were modeled, the average detection was  $153 \pm 3$  targets for  $CO_2$  prioritization ( $222 \pm 5$  for  $CH_4$  prioritization).

The observed differences between CO<sub>2</sub> and CH<sub>4</sub> detection rates across all scenarios can be attributed to the larger baseline inventory of CH<sub>4</sub> sources, which inherently increases the likelihood of detection under the same operational conditions. CH<sub>4</sub> sources, often associated with clustered sectors such as oil and gas production or landfills, present more frequent opportunities for detection within a limited swath width. By contrast, CO<sub>2</sub> sources, typically tied to larger industrial emitters like power plants and cement factories, tend to be more spatially dispersed, resulting in fewer targets per repeat cycle.

The seasonal trend in Figures 6 and 7 across scenarios also reveals a lower count of detected targets during the Northern Hemisphere winter months, as expected from real-world solar illumination conditions. Because targets with solar zenith angles above 70° are pre-filtered, the number of viable targets is much lower during the winter months in the Northern Hemisphere. This pattern further illustrates the uneven distribution of point emission sources globally (Figure 1, left panels), with the Southern Hemisphere's summer months contributing fewer additional targets due to the relatively low density of emission sources compared to the Northern Hemisphere (Figure 2).







Cloud cover remains a dominant challenge in satellite-based greenhouse gas (GHG) observations. According to Krijger et al. (2007), even relatively low cloud fractions significantly affect retrieval accuracy, often necessitating stringent cloud filtering that compromises data yields. This effect is exacerbated in tropical regions, as demonstrated by Frankenberg et al. (2024), where shallow cumulus clouds during the wet season limit valid measurements to as low as 0.1% of attempted observations. Hence, special attention must be directed toward resolving cloud-related challenges to improve the detection performance of systems like TANGO.

In our study, the Clear-Sky scenario provided an upper limit for TANGO's detection capability, free from cloud interference. However, the Cloud-Filtered scenario revealed a substantial reduction in detected targets due to global cloud coverage, with approximately 68% fewer detections in CO<sub>2</sub> prioritization and 64% in CH<sub>4</sub> prioritization. These reductions align closely with global cloud coverage statistics derived from MODIS cloud mask data that we used in our study, where the yearly average of "probably cloudy" and "confidently cloudy" conditions (flags in cloud mask data that were considered cloudy in the simulations) accounts for 70.65%, consistent with the findings of Stubenrauch et al. (2013). Given that much of the Earth's surface is persistently cloud-covered, strategies to tackle these obstacles are important for improving satellite detection yields.

One promising avenue for mitigating cloud interference is leveraging finer spatial resolution, as noted by Frankenberg et al. (2024). Their work suggests that high-resolution systems ( $\sim$ 200 m) can exploit gaps between clouds to achieve significantly higher observational yields, even in cloud-dense regions. With its proposed resolution of 300 m, TANGO is well-equipped to take advantage of such gaps, particularly in demanding environments like the humid tropics. Additionally, TANGO includes an exploratory operational mode with a resolution/sampling of 200 m, further enhancing its potential in adverse atmospheric conditions. However, finer resolution alone cannot fully overcome the limitations created by persistent cloud cover, necessitating additional strategies.

Our study investigated the integration of cloud forecast data as a complementary method to mitigate cloud interference. By incorporating dynamic forecast information, satellite systems can optimize target selection in near real-time, avoiding regions with high cloud cover and focusing on high-visibility targets. This cloud avoidance strategy, implemented through daily updates to the satellite's predefined target list, has the opportunity to strengthen detection yields. Simulations incorporating this approach demonstrated marked improvements: target detections increased by approximately 34.6% in CO<sub>2</sub> prioritization and 22.1% in CH<sub>4</sub> prioritization when compared to the Cloud-Filtered scenario. Even when forecast uncertainties were introduced in the Forecast +1d subset, the number of detected targets consistently surpassed those achieved with post-filtered static cloud masking.

These findings reinforce the utility of integrating forecast-based adaptive strategies into operational satellite systems. While cloud forecast accuracy and update frequency remain technical challenges, the demonstrated improvements in detection numbers present a robust case for their inclusion in future targeting satellite missions. In the context of TANGO, such an approach complies with its design as a flexible research satellite, capable of dynamically adjusting observation plans to maximize yield in varying atmospheric conditions.






While TDL represents the current detection limits of TANGO's operational framework, EDL illustrates the speculative future potential of next-generation targeting satellites with higher spatial resolution but similar orbital mechanics to TANGO. EDL scenarios expand observational capability by capturing smaller emitters that are undetectable under TDL. These sources, while increasing the total number of detected targets across all scenarios, may be associated with larger uncertainties in emission estimates due to their lower emission strengths. For instance, under Clear-Sky scenarios, CH<sub>4</sub> targets in Africa increase from 62 under TDL to 92.5 under EDL, while Asia's CH<sub>4</sub> targets rise from 206.1 to 303. Across all scenarios, the total number of detected targets often doubles under EDL compared to TDL, reflecting the broader reach of lower detection thresholds. Although these results are promising, they represent a hypothetical future capability and should be considered supplementary to the current mission. The primary focus remains on TDL, which already addresses the lion's share of global point source emitters.

Another method for augmenting the number of observable detections is deploying a constellation of targeting satellites akin to TANGO. Such a fleet of satellites could advance observational capabilities compared to single satellite systems by reducing revisit times and expanding coverage. Satellite constellations have already been applied in other missions. For instance, GHGSat employs a constellation of small satellites to monitor methane emissions, improving detection and source attribution from point sources (MacLean et al., 2024). Similarly, Carbon Mapper plans to deploy a multi-satellite constellation to monitor  $CO_2$  and  $CH_4$  emissions with high spatial resolution (Jacob et al., 2022).

While TANGO currently needs approximately four days for near-global coverage, a constellation of four similar systems in evenly spaced, phased orbits could achieve comparable coverage in one day. Our single-satellite simulations show TANGO detects about 500 targets per four-day repeat cycle under clear-sky conditions and about 120 under realistic cloud cover. A four-satellite constellation would therefore provide roughly 500 detections each day, effectively quadrupling the temporal yield without altering per-satellite performance. The higher revisit frequency would improve monitoring dynamic emission patterns, such as intermittent releases. Additionally, by increasing the number of satellites, the likelihood of capturing targets during clear-sky conditions or better exploiting gaps in cloud cover would improve significantly. Frequent revisit cycles would also facilitate more adaptable prioritization strategies, allowing dynamic reallocation of observation priorities to high-impact or under-monitored emitters.

The concept of a constellation is not without challenges, including the need for efficient management of multiple satellites, optimization of deployment strategies, and mitigation of risks associated with orbital congestion and debris (Curzi et al., 2020). The technical complexity of such coordination, along with the costs of launching and maintaining multiple satellites, cannot be overstated. Nevertheless, advancements in small satellite technology and the increasing availability of launch services suggest that a targeting constellation holds substantial promise for advancing global GHG monitoring.

# 6 Conclusions







In this study, we assessed the potential of targeting satellites, using the upcoming TANGO mission as a representative case, to monitor anthropogenic CO<sub>2</sub> and CH<sub>4</sub> point source emissions. By leveraging simulations based on TANGO's orbital parameters, detection limits, and dynamic maneuverability, we evaluated its capacity to observe major and minor emission point sources under varying scenarios, including Clear-Sky, Cloud-Filtered, and Cloud-Forecast conditions.

Our results show that TANGO has the potential to detect approximately 500 targets per repeat cycle under current detection limits (TDL), depending on the prioritization schemes used, covering a significant proportion of CO<sub>2</sub> and CH<sub>4</sub> point sources globally. Cloud coverage, however, emerged as a dominant factor influencing detection yields, with approximately 68% fewer detections for CO<sub>2</sub> prioritization in Cloud-Filtered scenarios. Integrating cloud forecast data effectively mitigated this limitation, improving detection yields by 34.6% for CO<sub>2</sub> and 22.1% for CH<sub>4</sub> prioritization. Over the full year, TANGO detects approximately 36,748 targets under CO<sub>2</sub> prioritization in the Clear-Sky scenario, dropping to 10,857 in the Cloud-Filtered scenario. Cloud-Forecast improves detection to 16,607, while the Forecast+1D scenario yields around 13,016 targets. Since CH<sub>4</sub> sources are more numerous than CO<sub>2</sub> point sources, overall detection counts are higher under CH<sub>4</sub> prioritization, following similar trends. These findings indicate that Cloud-Forecast represents an upper bound on forecast-driven target selection benefits, while Forecast+1D provides a more realistic lower bound. The difference between these cases shows that forecast inaccuracies significantly reduce detection gains, limiting the benefits of adaptive targeting.

In addition to the TDL simulations, we also looked into the potential of enhanced detection limits, which captured smaller emitters and showed promising prospects. Despite the capabilities of the current single-satellite design of TANGO, our study proposes the deployment of a satellite constellation. A constellation of TANGO-like satellites could significantly reduce revisit times, improve coverage, and enhance the adaptability of prioritization strategies, enabling more dynamic monitoring of GHG emissions.

In summary, targeting satellites like TANGO offer substantial potential in advancing global GHG monitoring that could complement existing systems and fill critical gaps in emission inventories. The integration of cloud forecast data enhances detection efficiency, while future progress in detection thresholds, spatial resolution, and satellite constellations will further strengthen their function in supporting international climate agreements and informing policies geared toward mitigating climate change.

Data availability. The point-source detection lists for the March 18–22, 2022 simulation (Clear-Sky, Cloud-Filtered, and Cloud-Forecast scenarios) are included in the Supplementary Material. The complete global point source data for greenhouse gas emission (GPS) used in this study was developed by TNO and made available for research purposes to the study team in support of the TANGO mission. For questions, future use, and use in other areas, please contact TNO (hugo.deniervandergon@tno.nl). The MOD35\_L2 cloud mask data from the MODIS instrument (DOI: 10.5067/MODIS/MOD35\_L2.061) are publicly accessible.

# **Appendix A: Continental Detection Statistics**

# A1 CH<sub>4</sub> prioritization Detection Counts

Table A1 provides the numerical breakdown of detected CH<sub>4</sub> and CO<sub>2</sub> sources across continents for different cloud scenarios under CH<sub>4</sub> prioritization for both TDL and EDL settings. The values represent the average number of detected sources per 4-day repeat cycle.

**Table A1.** Average number of CH<sub>4</sub> and CO<sub>2</sub> targets detected per four-day interval across continents under CH<sub>4</sub> prioritization for TDL and EDL detection limits.

| Continent     | Туре | Clear Sky       |        | Cloud-Filtered  |        | Cloud Forecast  |                 | Forecast +1d    |        |
|---------------|------|-----------------|--------|-----------------|--------|-----------------|-----------------|-----------------|--------|
|               |      | CH <sub>4</sub> | $CO_2$ | CH <sub>4</sub> | $CO_2$ | $\mathrm{CH}_4$ | $\mathrm{CO}_2$ | CH <sub>4</sub> | $CO_2$ |
| Africa        | TDL  | 62.0            | 9.0    | 23.2            | 5.3    | 26.0            | 6.3             | 23.4            | 5.7    |
|               | EDL  | 92.5            | 33.5   | 43.6            | 17.2   | 50.3            | 19.5            | 46.0            | 17.2   |
| Asia          | TDL  | 206.1           | 103.0  | 89.1            | 34.6   | 111.6           | 42.9            | 94.0            | 34.2   |
|               | EDL  | 303.0           | 241.5  | 122.8           | 72.6   | 150.2           | 97.8            | 127.9           | 77.7   |
| Europe        | TDL  | 43.3            | 32.6   | 10.8            | 7.3    | 14.0            | 11.5            | 10.3            | 8.0    |
|               | EDL  | 105.9           | 69.4   | 23.2            | 14.7   | 31.6            | 24.3            | 22.1            | 17.0   |
| North America | TDL  | 58.9            | 13.9   | 21.6            | 5.9    | 28.5            | 9.8             | 22.6            | 7.6    |
|               | EDL  | 143.0           | 36.9   | 44.8            | 13.3   | 65.4            | 23.4            | 50.3            | 18.5   |
| Oceania       | TDL  | 15.0            | 8.0    | 5.1             | 1.9    | 8.4             | 2.9             | 6.7             | 2.0    |
|               | EDL  | 24.0            | 18.0   | 9.6             | 5.5    | 16.8            | 6.8             | 13.3            | 5.1    |
| South America | TDL  | 25.9            | 4.0    | 7.5             | 1.2    | 9.0             | 2.2             | 7.4             | 1.6    |
|               | EDL  | 52.6            | 38.0   | 19.1            | 13.2   | 24.5            | 17.8            | 19.5            | 14.1   |

# A2 CO<sub>2</sub> prioritization Detection Counts

Table A2 presents the corresponding numerical breakdown for CO<sub>2</sub> prioritization, showing how the selection scheme affects the distribution of detected sources under different cloud scenarios.

Author contributions. HCA, JL, and AB contributed to the conceptualization and methodology development. HCA conducted the satellite simulation study with input from JL, using an orbit simulator developed with contributions from PV. SD provided the GPS inventory and related support. HCA performed the formal analysis and created the figures. HCA wrote the original draft, while all authors contributed to the interpretation of results and the review and editing of the manuscript.

**Table A2.** Average number of CH<sub>4</sub> and CO<sub>2</sub> targets detected per four-day interval across continents under CO<sub>2</sub> prioritization for TDL and EDL detection limits.

| Continent     | Туре | Clear Sky       |        | Cloud-Filtered  |                 | Cloud Forecast  |        | Forecast +1d    |        |
|---------------|------|-----------------|--------|-----------------|-----------------|-----------------|--------|-----------------|--------|
|               |      | CH <sub>4</sub> | $CO_2$ | CH <sub>4</sub> | $\mathrm{CO}_2$ | CH <sub>4</sub> | $CO_2$ | CH <sub>4</sub> | $CO_2$ |
| Africa        | TDL  | 45.0            | 11.0   | 14.3            | 6.3             | 17.7            | 7.7    | 15.8            | 6.9    |
|               | EDL  | 58.8            | 33.9   | 25.6            | 16.1            | 33.0            | 20.8   | 30.2            | 17.8   |
| Asia          | TDL  | 53.7            | 119.6  | 18.7            | 38.4            | 32.7            | 60.6   | 25.8            | 47.6   |
|               | EDL  | 58.5            | 312.7  | 17.6            | 93.1            | 39.6            | 138.8  | 31.2            | 108.2  |
| Europe        | TDL  | 20.2            | 35.8   | 3.9             | 9.4             | 6.2             | 13.9   | 4.4             | 10.0   |
|               | EDL  | 41.7            | 90.1   | 8.6             | 21.3            | 14.5            | 36.6   | 10.3            | 25.7   |
| North America | TDL  | 24.3            | 42.4   | 7.7             | 15.7            | 13.3            | 23.7   | 10.4            | 18.2   |
|               | EDL  | 79.1            | 108.3  | 28.8            | 42.2            | 41.8            | 58.8   | 32.2            | 46.8   |
| Oceania       | TDL  | 11.0            | 11.0   | 3.2             | 3.1             | 5.8             | 4.1    | 4.3             | 2.9    |
|               | EDL  | 20.0            | 26.0   | 6.0             | 9.1             | 11.7            | 11.2   | 8.7             | 8.7    |
| South America | TDL  | 17.9            | 11.0   | 4.4             | 3.8             | 5.2             | 4.9    | 4.0             | 3.7    |
|               | EDL  | 30.4            | 61.8   | 11.4            | 19.9            | 15.6            | 29.6   | 12.0            | 23.3   |

Competing interests. At least one of the (co-)authors is a member of the editorial board of Atmospheric Measurement Techniques.

# Disclaimer.


Acknowledgements. This work is supported by the AVENGERS project, funded by the European Union's Horizon Europe research and innovation programme under Grant Agreement No. 101081322, and the ITMS-BII-HISAT project within the ITMS – Integriertes Treibhausgas Monitoring System für Deutschland project, funded by the German Federal Ministry of Research, Technology and Space (BMFTR) under Grant Agreement No. 01LK2309A. The authors gratefully acknowledge the support of the state of Baden-Württemberg through bwHPC and the German Research Foundation (DFG) under grant INST 35/1597-1 FUGG. We also thank the data storage service SDS@hd, which is supported by the Ministry of Science, Research and the Arts Baden-Württemberg (MWK) and the DFG under grant INST 35/1503-1 FUGG.

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
