# Peer review of "Assessing the Detection Potential of Targeting Satellites for Global Greenhouse Gas Monitoring: Insights from TANGO Orbit Simulations"

_EGUsphere, 2025_

## Author Response (AR1)

**Manuscript: "Assessing the Detection Potential of Targeting Satellites for Global Greenhouse Gas Monitoring: Insights from TANGO Orbit Simulations"**

**by Charuvil Asokan et al.**

**Response to comment by anonymous referee #1**

**RC1:** Charuvil Asokan et al. present a simulation study on the detection coverage of point-source emitters from space, focusing exclusively on the TANGO mission concept and the impact of cloud cover on the observation potential, as well as the impact of using 1-day cloud forecasts for target scheduling. The manuscript is very well written, clearly structured and reads easily. The study falls roughly into the category of mission concepts studies as have been published in AMT previously (e.g, Butz et al., 10.5194/amt-8-4719-2015, or Edwards et al. 10.5194/amt-11-1061-2018) and thus fit well into the scope of the journal.

I see the manuscript mostly fit for publication, and have only cosmetic and minor suggestions to clarify certain points.

**Response:** We thank the reviewer for the positive evaluation and the constructive suggestions. We have revised the manuscript accordingly and will implement the required changes in the next iteration. Below, we detail how each point has been addressed in the revised version.

**1. Manuscript title**

**RC1:** "The manuscript title is certainly sufficient and informative, however I would suggest considering slightly changing the title to "[...]: Insights from TANGO Orbit Simulations" (which is also the way you describe it in the abstract). See my comment later, I believe it would help to clarify the context of the study for readers looking at the title."

**Response:** We agree with this suggestion. We have updated the title to "Assessing the Detection Potential of Targeting Satellites for Global Greenhouse Gas Monitoring: Insights from TANGO Orbit Simulations."

**2. Unit formatting**

**RC1**: "AMT requires units to be typed in exponential fashion, for example: kt year\$^{-1}\$ or kt per year, rather than kt / year. There are many occurrences in the text where unit typing should be corrected."

**Response**: All units have been revised to comply with AMT style (e.g., "kt year $^{-1}$ " and "Mt year $^{-1}$ ").

**3. Regarding Section 3**

**RC1**: "I believe it would be helpful to readers to insert a (short) paragraph that just quickly outlines the overall structure of the simulations before going into the details of each segment. Readers will have an easier time understanding the purpose of the simulations if they read about the overall flow, which seems to be: (1) point source list, (2) selecting the observation scheme (prioritization, forecast Y/N), (3) running orbital simulations that includes maneuvering and SZA flagging, (4) cloud filtering and finally (5) obtaining list of observed point sources to be compared against (1)."

**Response**: We have added the following overview at the start of Section 3 (Simulation Design and Methodology):

"In this section, we outline our simulation workflow. We start from the TNO Global Point Source inventory to list all CO\$\_2\$ and CH\$\_4\$ emitters. Next, we define observation schemes based on two detection thresholds (TDL and EDL) and two prioritization rules—one favoring CO\$\_2\$ sources and one favoring CH\$\_4\$ sources. Then, for each scheme, we run a four-day orbit simulation that includes TANGO's roll and pitch maneuvers and excludes overpasses with solar zenith angles above 70°. After that, we apply three cloud-treatment approaches: ideal clear-sky conditions, post-selection filtering using MODIS cloud masks, and pre-selection using cloud forecasts (with a one-day offset test). Finally, we compare, analyze, and discuss the detected targets under each scenario."

**4. "End-to-End simulator" terminology**

**RC1**: "P7 L160 There is mention of End-to-End simulation. In the context of an emission monitoring satellite, a true End-to-End system would start at emissions and end at some plume inversion (or similar), using the intermediate results from the retrieval process as well. I would suggest to change the term to "orbit simulator". I concede that "end-to-end" has some legitimacy due to the fact that you start with a point source inventory and end up with a subset of "observed" point sources (following the selection shown in Figure 4)."

**Response**: We have replaced all instances of "End-to-End simulator" with "orbit simulator" to avoid potential confusion. For example, in Section 3.1 now begins, "In this study, we employed an orbit simulator to generate satellite trajectories for TANGO."

**5. Data availability**

**RC1:** "My only main criticism relates to data availability. As per AMT's open data policy (https://www.atmospheric-measurement-techniques.net/policies/data\_policy.html), authors should attempt to at least provide the results used to produce final numbers and figures, and if that is not possible, provide a detailed statement as to why providing the results was not possible. The authors currently just state that the initial inventory is available on "the TNO FTP server" (no address given) upon request."

**Response:** We now provide the detection lists (geolocation, Emission strength, and Sector for selected CO2 and CH4 targets) for the March 18-22, 2022, four-day simulation run (Clear-Sky, Cloud-Filtered, and Cloud-Forecast scenarios) that we used to generate Fig. 5 as supplementary material. The revised data availability statement reads:

"The point-source detection lists for the March 18–22, 2022 simulation (Clear-Sky, Cloud-Filtered, and Cloud-Forecast scenarios) are included in the Supplementary Material. The complete global point source data for greenhouse gas emission (GPS) used in this study was developed by TNO and made available for research purposes to the study team in support of the TANGO mission. For questions, future use, and use in other areas, please contact TNO (hugo.deniervandergon@tno.nl). The MOD35\\_L2 cloud mask data from the MODIS instrument (DOI: 10.5067/MODIS/MOD35\\_L2.061) are publicly accessible."

We appreciate the reviewer's careful reading and helpful suggestions. We trust these changes improve clarity, style compliance, and data transparency.

- Harikrishnan Charuvil Asokan, on behalf of all co-authors

**Response to comment by anonymous referee #2**

RC2 Summary: Charuvil Asokan et al. present a sensitivity study that evaluates the detection coverage of point-source emitters from space, with a focus on the TANGO mission concept. The study explores how different observational scenarios (clear-sky, cloud-filtered, and cloud forecast) affect the potential for detecting CO2 and CH4 point source emitters. A key focus is the integration of one-day cloud forecast data into the targeting strategy, and how this improves observational efficiency under realistic atmospheric conditions. The authors examine two prioritization schemes: one that favors CH4 point sources over CO2, and another that prioritizes CO2. These schemes are evaluated under two detection capability scenarios, one representing current planned detection limits and one hypothetical future one with enhanced sensitivity. The study finds that integrating cloud forecast data improves detection yields, mitigating the limitations imposed by cloud cover. Specifically, the inclusion of forecast information increases detection efficiency by 34.6% when CO2 sources are prioritized and by 22.1% when CH4 sources are prioritized. Overall, the study provides valuable insight into how strategic observation planning and weather forecasting can be leveraged to enhance space-based detection of greenhouse gas point sources.

The manuscript is very well written and clearly structured. It fits well within the scope of the journal and makes a relevant contribution to the field of space-based greenhouse gas monitoring. I recommend publication after minor revisions.

**Response:** We thank the reviewer for their insightful feedback and positive evaluation. We have revised the manuscript accordingly and will implement the required changes in the final iteration. Below, please find our detailed responses to each reviewer's comment and the corresponding revisions made in the manuscript.

**Specific Comments:**

1. RC2: Page 9, Lines 213ff: The authors mention that at least 70% of clear-sky soundings within a 3km radius are sufficient for plume detection and that these thresholds were empirically selected. It would strengthen the manuscript to briefly elaborate on how these empirical thresholds were determined. For example, was this based on prior observational studies or model evaluation?

**Response:** Thank you for this suggestion. We have expanded Section 3.3 to explain that our two-step thresholds are informed by the findings in the literature (Krijger et al., 2007; Frankenberg et al., 2024). Specifically, the 70% / 30% cloud-cover cuts align with regimes where clear-scene yields increase sharply in these studies, and the 3 km check reflects TANGO's  $\sim$ 300 m pixel scale. This clarification should help readers understand the rationale for our cloud-filtering thresholds.

The updated paragraph in 3.3 reads,

"The cloud filtration process involves two steps, as depicted in Figure 4. After selecting the targets, in the first step, we assess the cloud coverage within a 50 km radius of each emission source. If the cloud cover exceeds 70%, the target is flagged as unsuitable for observation and excluded from further analysis. This coarse-scale screen balances discarding hopelessly overcast regions against retaining potentially observable scenes. Krijger et al. (2007) show that allowing up to 20% cloud at footprints of  $\sim$ 10000 km² raises clear-scene yield from  $\sim$ 3% (zero-cloud requirement) to  $\sim$ 17%. By analogy, our 70% cloud cut at 50 km excludes only the worst

30% of overcast areas while keeping most viable targets. For targets with less than 70% cloud cover, we proceed to the second filtering step: a localized 3 km-radius check that retains only those with  $\leq$  30% cloud cover (i.e.,  $\geq$  70% clear pixels). This finer check leverages TANGO's  $\sim$ 300 m pixels to exploit narrow "windows" through broken clouds. Frankenberg et al. (2024) demonstrate that  $\sim$ 200 m retrievals boost clear-scene frequency by 5–10× compared to kilometer-scale footprints. At 300 m resolution, a 3 km circle covers roughly  $\sim$ 300 independent pixels, so requiring  $\geq$  70% clear leaves more than 210 valid soundings per target. While this does not imply that retrievals can consistently produce a complete data product under such conditions, it provides sufficient clear pixels for robust plume detection. Only targets that pass both steps are considered viable, ensuring a high probability of visibility. These thresholds were chosen based on insights from the cited studies for simulation purposes and may not universally apply to all regions or cloud types, as persistent cloud formations, especially in the tropics, might require further refinement. By implementing this filtering mechanism, we better understand the influence of cloud cover and optimize the selection of realistic target numbers."

2. RC2: Page 10, Line 230ff: When introducing the Cloud-Forecast case, it would be helpful to provide information on the temporal requirements for incorporating forecast data into the target planning process. For example, do the forecasts need to be implemented 5-7 days in advance of target selection, or is a 1-2 day lead time sufficient? The authors mention later that the target list is updated daily. Including this operational detail earlier would improve clarity and help the reader better understand the practical implementation and constraints of the approach.

**Response:** We have added a brief statement at the start of Section 3.4 to specify how far in advance forecasts are assumed to be available and how often the target list is updated. The new text reads: "Operationally, we assume forecast fields for the upcoming cycle are available 24 h in advance, with daily updates to the target list."

**3. RC2:** Page 12, Figure 5: The meaning of the scatter point size is unclear. Does it represent emission strength, observation density, or another parameter? Please clarify this either in the figure caption or the main text.

**Response:** We have updated the Figure 5 caption to state that marker size corresponds to emission strength, scaled separately for CO2 and CH4.

**4. RC2:** Page 13, Figure 6: Please consider including the EDL case for the CO2 prioritization scenario here as well, similar to what is presented for CH4 in Figure 7. This addition would increase consistency across the figures and allow for a more direct comparison of detection performance under different scenarios. Further, since the manuscript discusses differences in the number of overpasses across prioritization schemes, it may be beneficial to place Figures 6 and 7 next to each other (e.g., as subpanels Figure 6a and 6b).

**Response:** We thank the reviewer for this suggestion. We believe it is best to maintain the original presentation and have added an additional figure to the Supplementary Material. We deliberately chose to show only the TDL results under CO2-prioritization in Figure 6 to introduce readers to the baseline performance of TANGO under its current detection limits. This keeps the initial parts of the results section uncluttered, allowing a clear comparison of monthly averages across the three simulation scenarios at TDL. In Figure 7, we then expand to both TDL and EDL under CH4-prioritization, enabling a direct side-by-side comparison of enhanced sensitivity.

Incorporating both TDL and EDL for  $CO_2$  in Figure 6 would disrupt this logical progression and risk overloading the reader early in the results. However, as it is essential to present the EDL results for the  $CO_2$ -prioritization scheme, we have added Supplementary Fig. S1, which replicates the format of Fig. 7 for  $CO_2$ -prioritization, showing monthly averages for both TDL and EDL across all scenarios. This approach preserves the pedagogical structure of the manuscript while providing the complete EDL comparison the reviewer requested.

**5. RC2:** Page 17, Lines 362–373: This paragraph provides a clear summary, but much of the content has already been presented in earlier sections. The same applies to several of the following paragraphs, which revisit previously discussed results. I suggest condensing this part of the discussion to improve the overall flow of the manuscript.

**Response:** We agree and have accordingly condensed the opening paragraph of the Discussion section to eliminate redundancy and have reduced repetitive content in subsequent paragraphs. Please see the revised manuscript for details.

**6. RC2**: Page 21, Line 510: Based on their findings, the authors propose the deployment of a satellite constellation to reduce revisit times, improve coverage, and enhance the adaptability of prioritization strategies. It would be helpful if the authors could elaborate on how their results support this proposal, especially considering that their orbit simulation and sensitivity study were conducted for a single instrument.

**Response:** Thank you for this suggestion. You are correct that we have not performed dedicated constellation simulations; rather, we offered a conceptual projection based on our single-satellite results. To clarify this link, we have expanded the discussion (end of Sect. 5) as follows:

"While TANGO currently needs approximately four days for near-global coverage, a constellation of four similar systems in evenly spaced, phased orbits could achieve comparable coverage in one day. Our single-satellite simulations show TANGO detects about 500 targets per four-day repeat cycle under clear-sky conditions and about 120 under realistic cloud cover. A four-satellite constellation would therefore provide roughly 500 detections each day, effectively quadrupling the temporal yield without altering per-satellite performance. The higher revisit frequency would improve monitoring dynamic emission patterns, such as intermittent releases. Additionally, by increasing the number of satellites, the likelihood of capturing targets during clear-sky conditions or better exploiting gaps in cloud cover would improve significantly. Frequent revisit cycles would also facilitate more adaptable prioritization strategies, allowing dynamic reallocation of observation priorities to high-impact or under-monitored emitters."

**RC2: Technical Corrections:**

- Page 1, Line 5: Change "good fraction" to "large fraction".
- Page 2, Line 25: Add "emissions" after "anthropogenic CO2 and CH4".
- Page 8, Title of Figure 3: Change "manoeuvre" to "maneuver" for consistency. While both spellings are correct, I suggest using consistent spelling throughout the manuscript.
- Page 11, Line 250: Change "realistic" to "near-realistic".
- Page 13, Line 279: Capitalize "Prioritization" at the beginning of the sentence.
- Page 21, Line 509: Remove "." at the end of the sentence.

**Response**: We appreciate the careful proofreading and have implemented all of these corrections.

We trust these revisions address the reviewer's concerns and improve the manuscript's clarity. Thank you again for your helpful suggestions.

- Harikrishnan Charuvil Asokan, on behalf of all co-authors